# (2,6-Dimethylphenyl)arsonic Acid Induces Apoptosis through the Mitochondrial Pathway, Downregulates XIAP, and Overcomes Multidrug Resistance to Cytostatic Drugs in Leukemia and Lymphoma Cells In Vitro

**DOI:** 10.3390/ijms25094713

**Published:** 2024-04-26

**Authors:** Nathalie Wilke, Corazon Frias, Albrecht Berkessel, Aram Prokop

**Affiliations:** 1Department of Pediatric Hematology/Oncology, Children’s Hospital Cologne, Amsterdamer Straße 59, 50735 Cologne, Germany; 2Department of Pediatric Oncology/Hematology, Helios Clinics Schwerin, Wismarsche Straße 393–397, 19049 Schwerin, Germany; 3Medical School Hamburg (MSH), University of Applied Sciences and Medical University, Am Kaiserkai 1, 20457 Hamburg, Germany; 4Department of Chemistry, Organic Chemistry, University of Cologne, Greinstraße 4, 50939 Cologne, Germany

**Keywords:** cancer chemotherapy, multidrug resistance, apoptosis, arsenic, leukemia, lymphoma, mitochondrial pathway, XIAP (x-linked inhibitor of apoptosis protein), synergy

## Abstract

Cancer treatment is greatly challenged by drug resistance, highlighting the need for novel drug discoveries. Here, we investigated novel organoarsenic compounds regarding their resistance-breaking and apoptosis-inducing properties in leukemia and lymphoma. Notably, the compound (2,6-dimethylphenyl)arsonic acid (As2) demonstrated significant inhibition of cell proliferation and induction of apoptosis in leukemia and lymphoma cells while sparing healthy leukocytes. As2 reached half of its maximum activity (AC50) against leukemia cells at around 6.3 µM. Further experiments showed that As2 overcomes multidrug resistance and sensitizes drug-resistant leukemia and lymphoma cell lines to treatments with the common cytostatic drugs vincristine, daunorubicin, and cytarabine at low micromolar concentrations. Mechanistic investigations of As2-mediated apoptosis involving FADD (FAS-associated death domain)-deficient or Smac (second mitochondria-derived activator of caspases)/DIABLO (direct IAP binding protein with low pI)-overexpressing cell lines, western blot analysis of caspase-9 cleavage, and measurements of mitochondrial membrane integrity identified the mitochondrial apoptosis pathway as the main mode of action. Downregulation of XIAP (x-linked inhibitor of apoptosis protein) and apoptosis induction independent of *Bcl-2* (B-cell lymphoma 2) and caspase-3 expression levels suggest the activation of additional apoptosis-promoting mechanisms. Due to the selective apoptosis induction, the synergistic effects with common anti-cancer drugs, and the ability to overcome multidrug resistance in vitro, As2 represents a promising candidate for further preclinical investigations with respect to refractory malignancies.

## 1. Introduction

One of the biggest challenges in cancer therapy are disease relapses due to emerging chemotherapy resistance [1,2]. The screening for novel potent anti-cancer drugs that can overcome resistance to established drugs and selectively induce apoptosis in malignant cells with low general or long-term toxicity remains crucial and is ongoing [3,4,5,6,7]. Metal- and metalloid-based compounds, like platinum-based cisplatin, have been successfully implemented as antiproliferative and apoptosis-inducing agents against malignant cells [8]. The effect on the redox system of mitotic cells is one of the main mechanisms of action of metal complexes to fight cancer cells [9]. The metalloid arsenic has already been used in the treatment of various diseases, like ulcers, malaria, and syphilis, for more than 2000 years [10], despite its proven toxic and carcinogenic properties [11,12,13]. Especially in hematologic malignancies like acute promyelocytic leukemia (APL), arsenic trioxide (As_2_O_3_) treatment has become the standard of care either in combination or after previous, unsuccessful treatment with all-trans retinoic acid (ATRA) [14,15,16,17]. Despite its great success rate in the treatment of APL and comparably low hematologic toxicity, As_2_O_3_ administration is still associated with a variety of side effects and toxicities affecting, e.g., the liver, the heart, or the neurological system, sometimes resulting in treatment discontinuation [18,19,20,21]. Similarly, other arsenic compounds like arsenic sulfide, darinaparsin, 4-(N-(S-glutathionylacetyl)amino)phenylarsenoxide, and arsenic hexaoxide have been candidates for the treatment of hematologic and solid cancers like lymphoma, melanoma, or pancreatic cancer [22,23,24,25,26]. Often, high doses and combinations with other drugs were necessary, giving rise to increased toxicity and decreased specificity [18]. A common anti-cancer mechanism of arsenic-based drugs is the induction of apoptosis by increasing reactive oxygen species and disrupting mitochondrial function [11,27,28,29,30,31]. Upregulation of pro-apoptotic genes and suppression of anti-apoptotic genes upon treatment have also been observed in cancer treatment with arsenic-based drugs [31,32,33,34].

In an effort to further reduce the dosage and thus toxicity of arsenic-based drugs in cancer treatment, the continuous development of novel drugs and delivery mechanisms is crucial.

Here, we investigated the arsenic compound (2,6-dimethylphenyl)arsonic acid, in short, As2, regarding its apoptosis-inducing and resistance-breaking properties in leukemia and lymphoma cell lines.

## 2. Results

### 2.1. As2 as a Unique and Biologically Active Candidate for Cancer Treatment

In initial screening experiments, the apoptosis induction by 11 novel arsenic compounds (As1–As11, Table 1) in Nalm-6 cells was investigated, covering a treatment range from 0.01 to 100 µM. The synthesis of the compounds was previously described [35]. Only As2 showed a promising effect on Nalm-6 cells in a low micromolar range compared to the other compounds, as is demonstrated by the respective AC50 concentrations, the concentrations that induced apoptosis in half of the cells calculated by nonlinear regression analysis (Table 1). Therefore, further analyses were focused on As2.

### 2.2. As2 Inhibits Proliferation and Induces Apoptosis in Cancer Cells In Vitro

Microscopic examination of Nalm-6 cells treated with different concentrations of As2 for 72 h (Figure 1a) revealed morphological differences from untreated cells and indicated As2-mediated inhibition of cell proliferation and induction of cell death. Compared to untreated and DMSO (dimethyl sulfoxide)-treated controls forming cell clusters of small, round cells, As2-treated cells show typical signs of apoptosis, such as swelling of cells followed by “blebbing” and cell fragmentation, as well as a continuous segregation of cell clusters into single cells. The morphological effects were dose-dependent and are visually recognizable starting at a concentration of 5 µM. At 20 µM, As2-induced effects on Nalm-6 cell viability and proliferation were morphologically comparable to Nalm-6 cells treated with an effective dose of 22.7 nM Vincristine (Vcr). The anti-proliferative impact of As2 was tested in Nalm-6 and BJAB mock cells by incubation with different As2 concentrations for 24 h and assessment of cell number and viability. Relative cell proliferation and proliferation inhibition for different As2 concentrations compared to untreated cells are depicted in Figure 1c. For both cancer cell types, proliferation is already impaired by 1 µM of As2. Starting from 5 µM for Nalm-6 and 1 µM for BJAB mock, As2 induces cell death, indicated by a reduction in cell concentration below the seeding concentration (1 × 10^5^ cells/mL). Unspecific necrotic cell death induced by As2 was excluded by photometric quantification of early LDH (lactate dehydrogenase) release in Nalm-6 cells treated with different concentrations of As2. LDH was not released into the supernatant by As2-treated cells, indicating membrane integrity and thus the viability of Nalm-6 cells after 1 h of incubation with As2 (Figure 1b). Apoptosis induction in Nalm-6 cells by As2 was examined and quantified by the detection of hypodiploid DNA as a result of DNA fragmentation occurring in the course of apoptosis. A screen for effective concentrations included As2 concentrations ranging from 0.01 to 100 µM. A dose-dependent increase in apoptotic Nalm-6 cells after incubation with As2 for 72 h was noticed (Figure 1d). The AC50 was calculated to be reached at circa 6.3 µM (Table 1), and the highest tested concentration (100 µM) resulted in approximately 80% of apoptotic cells. To confirm apoptosis induction by As2 and to distinguish between early and late apoptosis as well as necrosis in As2-treated Nalm-6 cells, an Annexin-V/propidium iodide (PI) assay was performed. Annexin-V-FLUOS stains cells in early and late apoptosis by binding to phosphatidylserine, which is translocated from the inside to the outside of the cell membrane during early apoptosis [36]. PI was used to mark cells with disrupted cell membranes in late apoptosis and necrosis. Nalm-6 cells that were subjected to 48 h of As2 treatment were dose-dependently induced to undergo apoptosis. Equal amounts of those cells were detected in early and late apoptosis (Figure 1e). Confirming the LDH assay, necrotic cells were not detected.

### 2.3. As2 Acts Selectively on Malignant Cells

To test the selectivity of As2 towards cancer cells, the ability to induce apoptosis was investigated simultaneously for Nalm-6 and BJAB mock cells as well as healthy human leukocytes. The analysis of DNA fragmentation in As2-treated cells revealed that for Nalm-6 and BJAB mock cells, a dose-dependently increasing fraction of cells underwent apoptosis upon As2 treatment (Figure 2). In contrast, only a small fraction (up to approximately 10%) of leukocytes were affected by all tested As2 concentrations. Significant differences in the effective induction of apoptosis in Nalm-6 and BJAB mock cells were detected for 5, 10, and 20 µM of As2, with initially more apoptotic Nalm-6 cells compared to BJAB mock cells at 5 and 10 µM, but a larger fraction of apoptotic BJAB mock cells at 20 µM. More leukocytes than Nalm-6 cells underwent apoptosis when treated with DMSO and 1 µM of As2. For higher concentrations starting at 5 µM of As2, the leukocytes were significantly less affected than Nalm-6 or BJAB mock cells.

### 2.4. As2 Overcomes Multidrug Resistance In Vitro

Multidrug resistance (MDR) is a widespread challenge in the therapy of malignant diseases and a huge contributor to disease relapse and poor patient prognosis. One common cause of multidrug resistance is a high expression of the multidrug transporter P-glycoprotein (P-gp), which enables increased exports of drugs out of the cancer cell and prevents successful treatment [37]. P-gp was found to be upregulated in the Vincristine- and Daunorubicin-resistant cell lines BeKa and LiKa, which is most likely the cause for the present multidrug resistance to several common drugs [38,39]. Drug resistance is depicted in Appendix A for all resistant cell lines used in this study. For the Vincristine- and Doxorubicin-resistant cell lines BiBo and 7CCA, the resistances are most likely triggered by the overexpression of the anti-apoptotic *Bcl-2* (B-cell lymphoma 2) and the downregulation of the effector caspase-3, respectively [40,41]. For the Daunorubicin-resistant cell line NiWi-Dau, a downregulation of the proapoptotic protein Harakiri (Hrk) was detected as the potential origin of drug resistance [42]. As2 was tested on drug-resistant cell lines as a potential resistance-breaking anti-cancer drug. For Nalm-6 cells resistant to Prednisolone (NaKu), Cytarabine (MaKo), Vincristine (BeKa), and Daunorubicin (LiKa), as well as K562 cells resistant to Daunorubicin (NiWi-Dau) (the co-resistances to other drugs are stated in Appendix A), As2 induced apoptosis in more than 50% of the cells at a concentration of 5 µM or higher (Figure 3). Up to approximately 90% apoptosis induction was reached. In most of the cases, As2 induced apoptosis in a significantly higher fraction of cells in the resistant cell lines compared to the respective control cell line without acquired resistance. The resistance established by exposure to increasing drug concentrations was confirmed by positive controls with effective doses of the respective drug.

### 2.5. As2 Increases Cancer Cell Sensitivity to Cytostatic Drugs

Combination chemotherapy is often used with the aim of reducing adverse effects by using lower doses of the single drugs and increasing treatment efficacy through synergistic effects. To evaluate the potential of As2 for combination chemotherapy, its ability to sensitize cancer cells to cytostatic drug treatment was investigated by co-treatment of Nalm-6 cells with low cytostatic doses of commonly used drugs and different concentrations of As2. The fraction of resulting apoptotic cells was used as the effective readout. Synergistic effects of As2 were observed in combination with Vincristine, Daunorubicin, and Cytarabine (Figure 4). As observed, 1 µM of As2 synergistically enhanced the effect of 8 nM of Daunorubicin more than fivefold (Figure 5b). Synergistic effects exceeding 100% were also detected for the combination treatments of Cytarabine and As2 (Figure 4c), highlighting the great synergistic potential of the compound As2 in combination treatments.

### 2.6. Disentangling As2-Induced Apoptosis Mechanisms

For the characterization of As2-induced apoptosis, cell models with modifications in apoptosis-related gene expression were analyzed in the context of As2 treatment.

#### 2.6.1. As2 Acts Independently of *Bcl-2* and Caspase-3 Expression Levels

In contrast to BJAB mock cells, the cell lines BiBo and 7CCA are characterized by an overexpression of the anti-apoptotic *Bcl-2* and an underexpression of caspase-3, a main executor of apoptosis, respectively. After treatment with As2, apoptosis induction in those cell lines was examined. As2 was able to induce apoptosis effectively in both modified cell lines (Figure 5a,b). This suggests that As2-induced apoptosis is neither dependent on caspase-3 nor on *Bcl-2* expression levels. Additionally, the significantly increased apoptosis induction in those cell lines compared to BJAB mock cells denotes another confirmation of resistance-overcoming properties, since BiBo and 7CCA cells are, among others, resistant to Vincristine and Doxorubicin, respectively. Resistance to Vincristine or Doxorubicin was verified by the respective drug treatment, which resulted in apoptosis induction in BJAB mock cells but not in the resistant cell lines.

#### 2.6.2. XIAP Downregulation by As2

Regulators of the apoptosis pathways were analyzed through western blot analysis in As2-treated cells. XIAP (x-linked inhibitor of apoptosis protein) is a member of the IAP (inhibitor of apoptosis protein) family and inhibits the effector caspase-3 and -7 as well as caspase-9 by binding [43,44]. XIAP protein expression was reduced in cells treated with Daunorubicin as well as in cells treated with 5 and 20 µM of As2, compared to controls and the treatment with 1 µM of As2 (Figure 5c). XIAP downregulation potentially caused reduced inhibition of caspase-3, -7, and -9 and promoted apoptosis in cells treated with As2.

#### 2.6.3. As2 Induces Apoptosis through the Mitochondrial Pathway

Upon induction of apoptosis, caspases, the apoptosis effector proteins, are activated by cleavage. In different apoptosis pathways, different caspases are the main initiators of the cell death program: caspase-8 in the extrinsic pathway and caspase-9 in the intrinsic pathway [45]. In As2-treated cells, the effector caspase-3 was shown to be processed and thus activated through western blot analysis (Figure 5c). However, the activation of caspase-3 was less prominent with increasing and more cytotoxic concentrations of As2. In the mitochondrial apoptosis pathway, caspase-9 acts upstream of caspase-3. Increased processing and activation of caspase-9 were detected in As2-treated Nalm-6 cells, indicating that the induced apoptosis is executed through the intrinsic mitochondrial pathway. With the aim of revealing or excluding a connection between As2 and the extrinsic apoptosis pathway, As2 was used to treat BJAB mock and BJAB FADDdn cells, and apoptosis was quantified through DNA fragmentation analysis. BJAB FADDdn cells harbor a dominant negative FADD (FAS-associated death domain) mutant, impairing the function of the FADD adapter protein, which is essential for signal transduction of extrinsic apoptosis induction through the Fas/CD95 receptor [46]. As2 induced apoptosis in the control line (BJAB mock) and the modified lymphoma cell line (BJAB FADDdn) (Figure 6a). The fraction of apoptotic cells was significantly higher in BJAB FADDdn cells treated with 5 and 10 µM of As2 compared to BJAB mock cells. A functional FADD adapter protein was not necessary for apoptosis induction by As2, supporting the hypothesis that As2 triggers the intrinsic rather than the extrinsic apoptosis pathway. In the mitochondrial apoptosis pathway, the proapoptotic protein Smac (second mitochondria-derived activator of caspases)/DIABLO (direct IAP binding protein with low pI) is translocated from the mitochondrium into the cytosol and binds to IAPs like XIAP, which interferes with their function of inhibiting caspase activity and thus enhances apoptosis execution. The T cell leukemia cell line Jurkat smac overexpresses Smac/DIABLO. Apoptosis induction by As2 was investigated in Jurkat smac cells and the control cell line Jurkat neo. In all tested concentrations, As2 induced significantly more apoptosis in Jurkat smac cells, whereas Jurkat neo cells were not sensitive to As2 treatment (Figure 6b). This indicates that As2-induced apoptosis through the mitochondrial pathway can be influenced by an excess of Smac/DIABLO. To further investigate the mitochondrial functionality and its association with apoptosis induction by As2, the mitochondrial membrane integrity, which is disrupted in the course of the intrinsic apoptosis pathway, was investigated by JC-1 staining for reduced mitochondrial membrane potential after 48 h of As2 treatment. While the cells in the controls and the cells treated with 1 µM of As2 did not lose mitochondrial membrane integrity, the fraction of cells with reduced membrane potential increased dose-dependently to approximately 50% of cells treated with 10 or 20 µM of As2 (Figure 6c). The insensitivity of As2-mediated apoptosis to FADD functionality, increased apoptosis induction in Smac/DIABLO-overexpressing cells, and loss of mitochondrial membrane integrity indicate that As2 acts through induction of the mitochondrial apoptosis pathway.

## 3. Discussion

Due to disease relapses and the occurrence of resistances to established cytostatic drugs during cancer therapy, screenings for new drug compounds that selectively target the malignant cells, inhibit cell proliferation, and induce programmed cell death are essential.

In cancer therapy, several metal- or metalloid-based drugs have been tested and successfully implemented in treatment regimens. Platinum-based drugs like cisplatin or carboplatin belong to the most common cytostatic drugs used in cancer therapy [8]. Yet, treatment success is often restricted by therapy resistance and adverse effects of the drug treatment [4,5]. Therefore, novel metal complexes are assessed as potential drug candidates for application in resistant tumors [7,8]. Previous research has investigated several novel drug compounds, among others based on gold, ruthenium, and cobalt, that showed promising antiproliferative and apoptosis-inducing characteristics in vitro in several drug-resistant cell lines [42,47,48,49]. Despite the toxic properties of the metalloid arsenic, different arsenic compounds have been used repeatedly in human medicine [10,24,50,51]. In cancer therapy, arsenic trioxide is actively used to treat all-trans retinoic acid-resistant acute promyelocytic leukemia (APL). Its mechanisms of action in leukemic cells are described, among others, as induction of differentiation through impact on APL fusion gene products, the induction of apoptosis, e.g., through reactive oxygen species that influence the redox system, DNA damage and activation of the mitochondrial apoptosis pathway, as well as anti-angiogenic properties [51,52,53,54,55]. Several other arsenic compounds have been examined for potential use in the treatment of various malignancies, especially in the context of hematologic malignancies [56]. In solid tumors, arsenic trioxide has not been successfully applied in vivo due to renal clearance, a short half-life, and organ toxicity. A current approach to circumventing these limitations is, for example, to perform drug delivery by nanotechnology [55].

This study investigated the potential of the novel arsenic compound (2,6-dimethylphenyl)arsonic acid (As2) to inhibit cancer cell proliferation and induce apoptosis in therapy-resistant cancer cell lines in vitro. The organic compound As2 was shown to inhibit proliferation and selectively induce apoptosis in leukemia and lymphoma cells without affecting healthy leukocytes. Furthermore, As2 was able to overcome multiple drug resistances.

With the aim of reducing the potential toxicities of cytostatic drugs, they are often combined at low doses in polychemotherapy [57,58,59]. In combination with common cytostatic drugs, As2 exhibited synergistic effects, lowering the required concentrations of both drugs in a combination therapy. The low effective apoptosis-inducing concentration, the selectivity towards malignant cells, and the synergistic potential for use in combination therapy are promising qualities of As2 for its application in patients, especially with regard to therapy-resistant relapses and reduction in adverse effects.

In line with previously described arsenic drugs, As2-induced apoptosis is likely executed through the mitochondrial apoptosis pathway, as indicated by the loss of mitochondrial membrane potential, activation of caspase-9, insensitivity towards a lacking functional FADD adapter protein, and the rescue of apoptotic activity by Smac/DIABLO overexpression in T cell leukemia cells. This is consistent with the findings that the main targets of arsenic-based compounds are mitochondrial proteins that regulate cellular redox states, mitochondrial respiration, and the production of reactive oxygen species (ROS). Targeting of those proteins, like thioredoxins, was shown to increase ROS and cellular stress, activate Ask1 signaling, and disrupt the regulation of the mitochondrial permeability transition pore, followed by cytochrome C release, thus eventually leading to apoptosis induction [9,60]. Recently, another novel organoarsenic compound was found that acts as an inhibitor of thioredoxin reductase, leading to the apoptosis of cancer cells through the mitochondrial pathway induced by oxidative stress [28].

However, successful apoptosis induction by As2 was found to not be dependent on caspase-3 and *Bcl-2* expression levels. In addition to caspase-3-mediated and caspase-7-mediated apoptosis, caspase-independent apoptosis execution might be activated by the apoptosis-inducing factor (AIF) protein. Further characterization of the apoptosis induction revealed that XIAP, a member of the inhibitor of apoptosis protein (IAP) family and inhibitor of caspase activity, is downregulated upon As2 administration. In combination with Smac/DIABLO activity, initiated during the mitochondrial apoptosis pathways, and counteracting caspase inhibition by blocking IAP activity [61,62,63], apoptosis execution by effector caspases is enforced. XIAP downregulation and Smac mimetics were found to sensitize acute lymphoblastic leukemia (ALL) cells towards chemotherapy and reduce tumor outgrowth in vivo [64]. We found that As2 activates the mitochondrial apoptosis pathway, downregulates XIAP, breaks chemotherapy resistance, acts independently of *Bcl-2* expression levels, and shows synergistic effects with common cytostatic drugs (Figure 7). Thus, As2 yields similar results in sensitizing leukemia cells to chemotherapy, as, for instance, small-molecule XIAP inhibitors that showed results against acute leukemia cells with *Bcl-2*-mediated resistance, synergizing with TRAIL (TNF-related apoptosis-inducing ligand) or Fas activation [65,66]. In human lymphatic endothelial cells, Hrgovic et al. recently found that arsenic trioxide induces both the intrinsic and extrinsic apoptosis pathways and downregulates the anti-apoptotic proteins XIAP, Survivin, and cIAP-2 [31].

In a previous study, As2 as well as four more of the 11 organoarsenic compounds screened in this study also showed promising antiplasmodial activity [35], thus representing a versatile group of new compounds for potential use in therapy against diseases like malaria or cancer.

Further studies to detangle the mechanisms of action, e.g., investigating the impact of As2 treatment on the redox system and the expression of pro- and anti-apoptotic proteins, as well as extensive toxicity studies, are warranted.

## 4. Materials and Methods

### 4.1. Chemicals and Drugs

Propidium iodide (50 µg/mL) and RNase A were obtained from Serva (Heidelberg, Germany) and Qiagen (Hilden, Germany), respectively. Common cytostatics were provided by the Children’s Hospital Cologne Amsterdamer Straße and the pharmacy of the Hospital Cologne Merheim, Kliniken der Stadt Köln (Cologne, Germany). The molecular structures of the investigated organoarsenic compounds As1 to As11 are listed in Table 1. The synthesis of these materials was described previously [18]. All drugs and compounds were dissolved in DMSO (Serva, Heidelberg, Germany) prior to the treatment of different cell lines.

### 4.2. Cell Lines and Cell Culture

Nalm-6 cells, human B cell precursor leukemia cells, were provided by Dr. K.-H. Seeger (Charité, Berlin, Germany). The BJAB cell lines (Burkitt-like lymphoma) containing a pcDNA4 control vector (BJAB mock) or transfected with the vector pcDNA3-FADD-dn (BJAB FADDdn) to express a dominant negative FADD mutant lacking the N-terminal death effector domain were obtained from Prof. Dr. P.T. Daniel (Charité, Berlin, Germany). Jurkat cell lines, derived from human T cell leukemia, were provided by Prof. Dr. Fulda (Frankfurt, Germany). Jurkat neo cells were transfected with a control vector, and Jurkat smac cells were transfected to overexpress the proapoptotic factor Smac/DIABLO. K562 cells, chronic myelogenous leukemia cells, were kindly provided by Dr. K.-H. Seeger (Charité, Berlin, Germany). Nalm-6, BJAB, and K562 cell lines resistant to different cytostatic drugs were established in our lab by exposure to increasing drug concentrations. NaKu, MaKo, BeKa, and LiKa cells were derived from Nalm-6 cells and were resistant to Prednisolone, Cytarabine (Ara-C), Vincristine, and Daunorubicin, respectively. The resistance of BeKa and LiKa cells to Vincristine and Daunorubicin is associated with the overexpression of P-glycoprotein 1 (P-gp) [19,20]. BiBo and 7CCA cells were generated from the BJAB cell line and acquired resistance to Vincristine and Doxorubicin, respectively. BiBo cells overexpress *Bcl-2* [21], whereas caspase-3 is underexpressed in 7CCA cells [22]. Additionally, K562 cells resistant to Daunorubicin (NiWi-Dau) were generated. The co-resistance of the mentioned cell lines to further cytostatic drugs was tested and is depicted in Appendix A. Human leukocytes were obtained from blood that was personally donated by the co-author, Prof. Dr. Aram Prokop. The cancer cell lines were cultivated in RPMI 1640 medium (Gibco, Invitrogen, Karlsruhe, Germany) supplemented with 10% FCS, 1% penicillin/streptomycin, and 1% L-glutamine at 37 °C with a CO_2_ content of 5%. Human leukocytes were grown in RPMI 1640 medium (Gibco, Invitrogen, Karlsruhe, Germany) supplemented with 20% FCS, 1% penicillin/streptomycin, and 1% L-glutamine. Cells were passaged 2–3 times per week. The cell concentration was adjusted to 3 × 10^5^ cells/mL to maintain standardized growth conditions 24 h prior to the experiments.

### 4.3. Determination of Cell Viability

To assess the viability and proliferation capacity of cancer cells treated with different concentrations of As2, viable cells were counted 24 h after treatment using the CASY^®^ Cell Counter and Analyzer System (OMNI Life Science, Bremen, Germany). Cells were initially seeded at a concentration of 1 × 10^5^ cells/mL into 6-well plates and treated with different concentrations of As2. DMSO-treated and untreated controls were included. After 24 h of incubation at 37 °C, 100 µL of cell suspension from each well was diluted in 10 mL of CASY^®^ton (ready-to-use isotonic buffer) and used for automated cell counting by the CASY^®^ Cell Counter. Following the normalization to the untreated negative control, the relative cell proliferation as well as the compound-induced proliferation inhibition for each condition were calculated.

### 4.4. LDH Release Assay

In order to characterize the cell death induced by As2, early lactate dehydrogenase (LDH) release into the supernatant as a sign of unspecific necrotic cell death was measured. Cancer cells were seeded in 6-well plates at a concentration of 1 × 10^5^ cells/mL and incubated at 37 °C and 5% CO_2_ with different concentrations of As2. DMSO and untreated negative controls were included. After 1 h, 100 µL of cell suspension from each well was transferred to a 96-well plate. As a positive control for LDH release due to cell lysis, previously untreated cells were treated with 0.1% TritonX-100 and incubated for 5 min at 37 °C. Cell culture medium was included as a control for background signals. Following centrifugation (1500 rpm, 5 min, 4 °C), the supernatant was diluted 1:2.5 in 1× PBS. LDH activity was detected using the Cytotoxicity Detection KitPLUS (LDH) (Roche, Basel, Switzerland) and the Multiskan Ascent 354 microplate reader (Thermo Scientific, Waltham, MA, USA) according to the manufacturer’s instructions. Absorbance at 492 nm was detected as a quantitative measure of the enzymatic activity. After subtraction of the background signal, measured values were normalized to the positive control, representing 100% cytotoxicity, and the negative control, representing 100% viability, in order to determine relative cytotoxicity and viability, respectively. Calculated relative viability values > 100% were defined as 100% viability.

### 4.5. Isolation of Human Leukocytes

In order to obtain human leukocytes, more specifically peripheral blood monocytes (PBMCs), blood samples personally donated by the co-author, Prof. Dr. Aram Prokop, were subjected to density gradient centrifugation. First, the blood was diluted 1:1.2 with RPMI 1640 medium (Gibco, Invitrogen, Karlsruhe, Germany) supplemented with 20% FCS, and 5 mL of the diluted blood sample was carefully layered on top of 4 mL of Ficoll separating solution (Saccharose-Epichlorhydrin-Copolymer, Merck, Darmstadt, Germany). After centrifugation (18 min, 18 °C, 2000 rpm), the layer of PBMCs between blood plasma and erythrocytes was collected and washed in RPMI 1640 medium (20% FCS). The PBMCs were collected by centrifugation (5 min, 18 °C, 2000 rpm) and resuspended in RPMI 1640 medium (20% FCS) for the determination of cell number and viability using the CASY^®^ Cell Counter prior to the experiment.

### 4.6. Western Blot Analysis

For the characterization of the induced apoptosis pathway, expression of apoptosis-related proteins was assessed by western blot analysis of Nalm-6 cells treated with DMSO, different concentrations of As2, and untreated negative controls, as well as positive controls treated with 76 nM of Daunorubicin after 36 h of incubation. First, the cells were washed twice with 1× PBS. After cell harvesting, cell pellets were lysed in lysis buffer containing 10 mM of Tris/HCl, pH 7.5, 300 mM of NaCl, 1% Triton X-100, 2 mM of MgCl_2_, 5 μM of EDTA, and 1 protease inhibitor cocktail tablet (Roche, Basel, Switzerland). Protein concentrations in the samples were determined with the BCA Protein Assay Kit (Thermo Fisher Scientific, Waltham, MA, USA) according to the manufacturer’s instructions and quantified using the Multiskan Ascent 354 microplate reader (Thermo Scientific, Waltham, MA, USA). Protein samples in equal amounts (40 mg per sample) were loaded onto a polyacrylamide gel (Serva, Heidelberg, Germany) and separated by SDS/PAGE. Proteins were transferred from the gel to a nitrocellulose membrane (Schleicher and Schuell, Dassel, Germany) using a Trans-Blot^®^ SD Semi-Dry Transfer Cell (Bio-Rad, Hercules, CA, USA). The membrane was blocked with 5% BSA in 1× PBST for 1 h at RT and incubated with the primary antibodies (Mouse/Rabbit anti-caspase-3/caspase-9/XIAP/β-actin; Santa Cruz Biotechnology, Dallas, TX, USA, and Sigma-Aldrich, St. Louis, MO, USA) for 1 h. The membrane was washed with 1× PBST at least three times for 15 min and then incubated with HRP-conjugated secondary antibodies (anti-mouse IgG HRP, Bioscience, San Diego, USA, and anti-rabbit IgG HRP, Promega, Minneapolis, MN, USA) for 1 h. After washing three times with 1× PBST, protein signals were detected using the enhanced chemiluminescence (ECL) system (Amersham Buchler, Braunschweig, Germany). Images were processed with ImageJ (Version 1.54g).

### 4.7. Measurement of DNA Fragmentation

Apoptotic cell death of cancer cells or leukocytes induced by the tested compounds and cytostatic drugs was evaluated by modified cell cycle analysis by determining the fraction of apoptotic cells after incubation with different concentrations of the respective compound. DNA fragmentation as a late event in apoptosis was detected on a single cell level by flow cytometry. Cells seeded at a concentration of 1 × 10^5^ cells/mL were treated with different concentrations of the compounds of interest, equivalent volumes of DMSO, different cytostatic drugs, or left untreated for 72 h at 37 °C and 5% CO_2_. Cells were collected by centrifugation (5 min, 8000 rpm, 4 °C) and fixed with 2% (*v*/*v*) formaldehyde in 1× PBS for 30 min. The fixed cells were collected by centrifugation and incubated for 15 min with ethanol/1× PBS (1:2, *v*/*v*). After centrifugation, the cells were resuspended in 1× PBS supplemented with 40 µg/mL of RNase A. After 30 min of incubation at 37 °C for RNA degradation, cells were pelleted by centrifugation and resuspended in 1× PBS containing 50 µg/mL of propidium iodide. Hypoploidy as a result of DNA fragmentation was flow cytometrically determined by a modified cell cycle analysis [23,24]. The fraction of hypoploid apoptotic cells was quantified using FACSCalibur^TM^ with CellQuest Pro Software (Version 6.1; Becton Dickinson, Heidelberg, Germany). Percentages of apoptotic cells detected as a percentage of hypoploidy (subG1 phase) for every sample were corrected for unspecific apoptosis in untreated cells.

### 4.8. Annexin-V/Propidium Iodide Apoptosis Assay

For the detection of As2-induced cell death and the differentiation between early or late apoptosis and necrosis, cells seeded at a concentration of 1 × 10^5^ cells/mL were treated with different concentrations of As2 for 48 h at 37 °C and 5% CO_2_. Cells were collected by centrifugation (5 min, 8000 rpm, 4 °C), washed with 1× PBS, and collected (5 min, 1500 rpm, RT) for double staining with Annexin-V-FLUOS and propidium iodide. The Annexin-V-FLUOS Staining Kit (Roche, Mannheim, Germany) was used for sample preparation, following the manufacturer’s instructions. Flow cytometric analysis was performed using FACSCalibur^TM^ with CellQuest Pro Software (Version 6.1; Becton Dickinson, Heidelberg, Germany). Viable cells were quantified as the cell fraction negative for both Annexin V staining and propidium iodide. Early apoptotic cells are defined as Annexin-V-positive and PI-negative. Late apoptotic and necrotic cells are represented by the double-positive and Annexin-V-negative/PI-positive fractions, respectively.

### 4.9. Measurement of Mitochondrial Membrane Permeabilization

The mitochondrial membrane permeability transition was measured as an indicator of As2-induced apoptosis through the mitochondrial pathway. Cells seeded at a concentration of 1 × 10^5^ cells/mL were incubated with As2 for 48 h at 37 °C and 5% CO_2_, collected (5 min, 3000 rpm, 4 °C), and resuspended in phenol red-free RPMI 1640 medium. JC-1 dye (5,5′,6,6′-tetrachloro-1,1′,3,3′-tetraethyl-benzimidazolyl-carbocyanin iodide, Molecular Probes, Leiden, The Netherlands) was added to the samples for a final concentration of 2.5 µg/µL. An unstained control was included. The samples were incubated at 37 °C for 30 min with moderate shaking. After harvesting the cells by centrifugation (5 min, 4000 rpm, 4 °C), they were resuspended in 1× PBS. Cells with decreased mitochondrial membrane potential (∆ψm) were quantified by flow cytometry using FACSCalibur^TM^ with CellQuest Pro Software (Version 4; Becton Dickinson, Heidelberg, Germany). Percentages of cells with reduced ∆ψm are given after subtraction of background control (untreated cells) signals and represent cells undergoing mitochondria-dependent apoptosis.

### 4.10. Statistical Analysis

Data are shown as the mean values of 3 replicates ± standard deviation. Microsoft Excel and GraphPad Prism (version 9.2.0) were used for calculations and statistical analyses. For parametric data, two-tailed t-tests or two-way ANOVA with post-hoc Bonferroni’s multiple comparisons test were used for evaluating statistical significance, as stated in the figure legends. Significance levels were determined as follows: significant (*) for *p* ≤ 0.05, highly significant (**) for *p* ≤ 0.01, and extremely significant (***) for *p* ≤ 0.001. For simplicity reasons, only significant comparisons are indicated in the figures. Synergistic effects were calculated using the concept of fractional products [67].

## 5. Conclusions

In summary, the arsenic compound As2 represents a new and promising metalloid-based anti-cancer agent with antiproliferative and apoptosis-inducing properties that is selectively effective against malignant cells at low micromolar concentrations. It could potentially be used in combination therapy with established cytostatic drugs and for application in patients with therapy-resistant cancer entities.

## Figures and Tables

**Figure 1 ijms-25-04713-f001:**
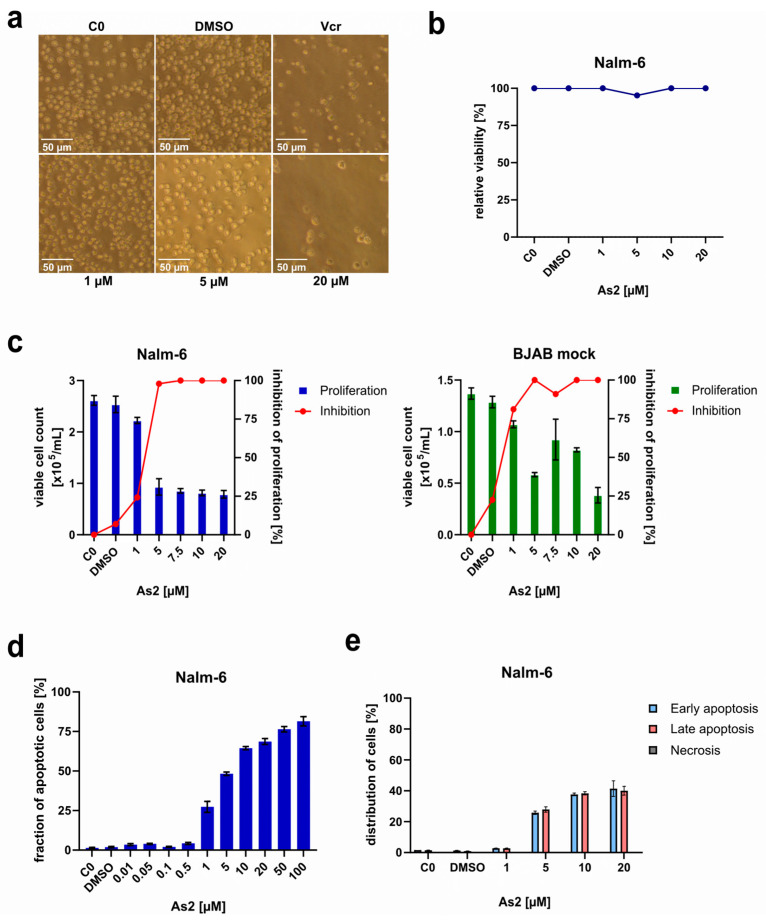
As2 treatment inhibits the proliferation of leukemia and lymphoma cells and successfully induces apoptotic cell death in vitro. (**a**) Nalm-6 cells were treated with DMSO (dimethyl sulfoxide), 22.7 nM of Vincristine (Vcr), and 1, 5, and 20 µM of As2, or left untreated. Images were taken after 72 h of incubation. (**b**) Nalm-6 cells were treated with different concentrations of As2 [µM], DMSO, or left untreated (C0) for 1 h. By quantifying levels of lactate dehydrogenase (LDH) in the supernatants relative to blank and lysed controls, cell viability was determined as a measure of necrotic cell death. The mean values of 3 replicates are shown. (**c**) Nalm-6 (**left panel**) and BJAB mock cells (**right panel**) were seeded at a concentration of 1 × 10^5^ cells/mL and treated with different concentrations of As2 [µM], DMSO, or left untreated (C0). After 24 h of incubation, cell counts [cells/mL] were determined (left *y*-axis) to calculate proliferation inhibition by As2 in percentages (right *y*-axis). The mean values of 3 replicates with a standard deviation (SD) are given for cell counts. The inhibition is shown as mean values. (**d**) To screen for apoptosis-inducing concentrations of As2, Nalm-6 cells were incubated for 72 h with As2 concentrations between 0.01 and 100 µM, DMSO, or left untreated (C0). The fraction of apoptotic cells in percentages was determined by flow cytometry, and the mean values with an SD of 3 replicates are depicted. (**e**) Annexin-V/Propidium iodide (PI) double staining and flow cytometric analysis of Nalm-6 cells treated with DMSO and different As2 concentrations [µM] for 48 h or left untreated (C0) was performed to evaluate the fraction of cells undergoing early and late apoptosis or necrosis. Fractions of cells in percentages are shown as mean values with an SD of 3 replicates.

**Figure 2 ijms-25-04713-f002:**
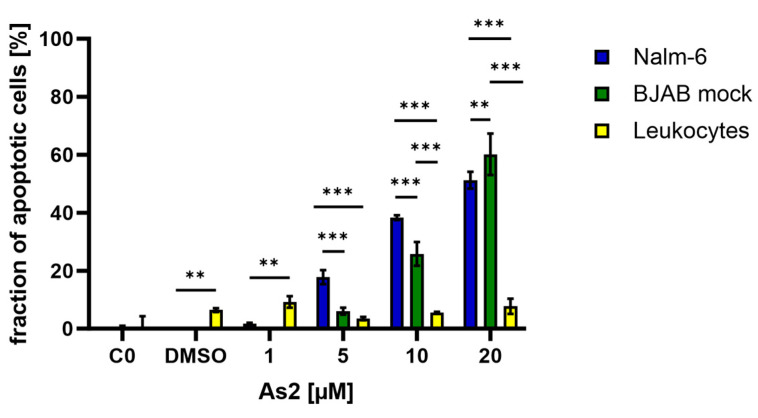
As2 selectively affects malignant cells. Nalm-6 and BJAB mock cancer cells and healthy leukocytes were treated with DMSO (dimethyl sulfoxide) or different concentrations of As2 [µM] and incubated for 72 h. Untreated negative controls (C0) were included. The fraction of apoptotic cells in percentages was determined by flow cytometry, and the mean values with an SD of 3 replicates are depicted. Significant differences between cell types for each concentration were determined by a two-way analysis of variance (ANOVA) with a post-hoc Tukey’s multiple comparison test. Significance levels are determined as follows: (**) for *p* ≤ 0.01, and (***) for *p* ≤ 0.001. For simplicity, only significant differences in the comparison between the healthy leukocytes and the respective cancer cell line responses are indicated.

**Figure 3 ijms-25-04713-f003:**
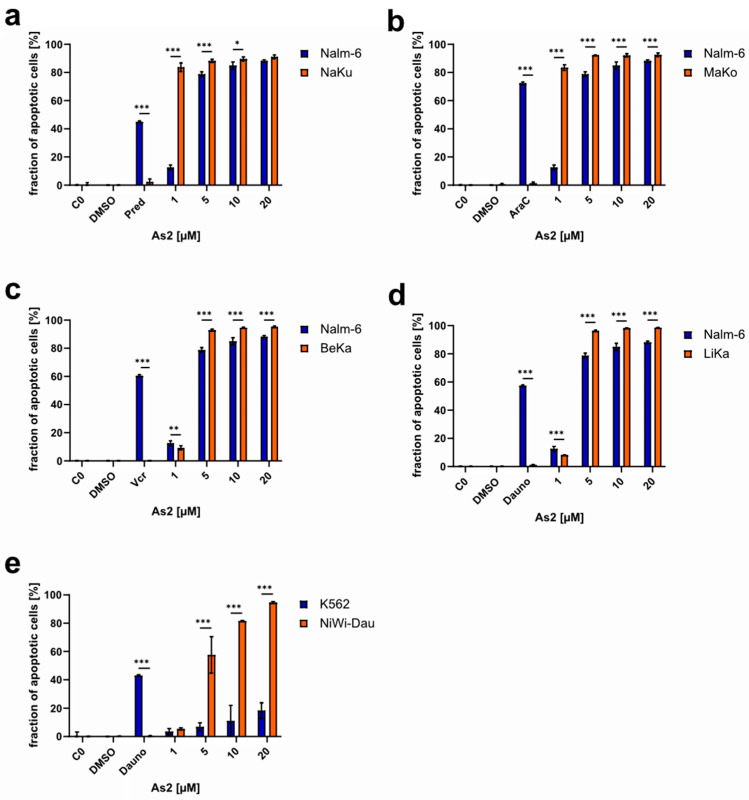
As2 overcomes multiple resistances to common cytostatic drugs by inducing apoptosis in resistant cell lines in vitro. Cell lines with established resistance to (**a**) Prednisolone (Pred; NaKu), (**b**) Cytarabine (AraC; MaKo), (**c**) Vincristine (Vcr; BeKa), and (**d**,**e**) Daunorubicin (Dauno; LiKa, NiWi-Dau) and the respective non-resistant parental cell line (Nalm-6, K562) were treated with different concentrations of As2 [µM] or DMSO (dimethyl sulfoxide) and incubated for 72 h. Untreated negative controls (C0) were included as well as positive controls for drug resistance. For positive controls, cells were treated with 55.6 nM of Prednisolone, 1.4 nM of Cytarabine, 22.7 nM of Vincristine, and 56.6 nM (LiKa/Nalm-6) or 2 µM (NiWi-Dau/K562) of Daunorubicin. The fraction of apoptotic cells in percentages was determined by flow cytometry, and the mean values with an SD of 3 replicates are depicted. Significant differences between cell types for each concentration were determined by a two-way analysis of variance (ANOVA) followed by Bonferroni’s multiple comparison test. Significance levels are determined as follows: (*) for *p* ≤ 0.05, (**) for *p* ≤ 0.01, and (***) for *p* ≤ 0.001. For simplicity, only significant differences are indicated.

**Figure 4 ijms-25-04713-f004:**
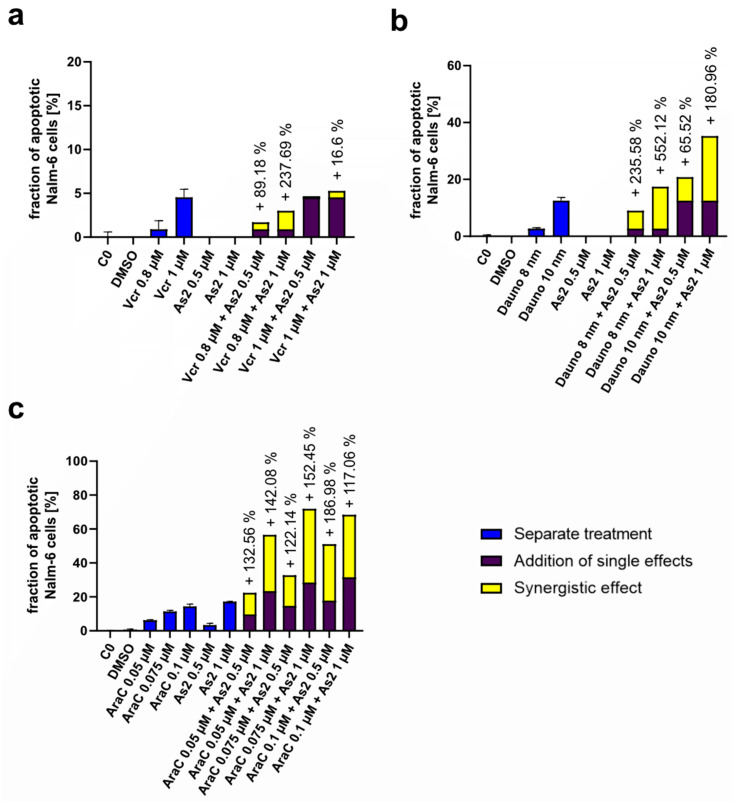
As2 sensitizes leukemia cells to common cytostatic drugs. Nalm-6 cells were treated with low doses of common cytostatic drugs and 1 or 2 µM of As2, DMSO (dimethyl sulfoxide), combinations of As2 and drug concentrations, or left untreated (C0) for 72 h. Synergistic effects of As2 with (**a**) Vincristine (Vcr), (**b**) Daunorubicin (Dauno), and (**c**) Cytarabine (AraC) were calculated after the flow cytometric analysis of apoptotic cells for each condition [%]. For single treatment conditions, the mean values with an SD of 3 replicates are given, and for combination treatments, the mean values of 3 replicates without an SD are given. Additive synergistic effects are depicted in percentages above the respective bars.

**Figure 5 ijms-25-04713-f005:**
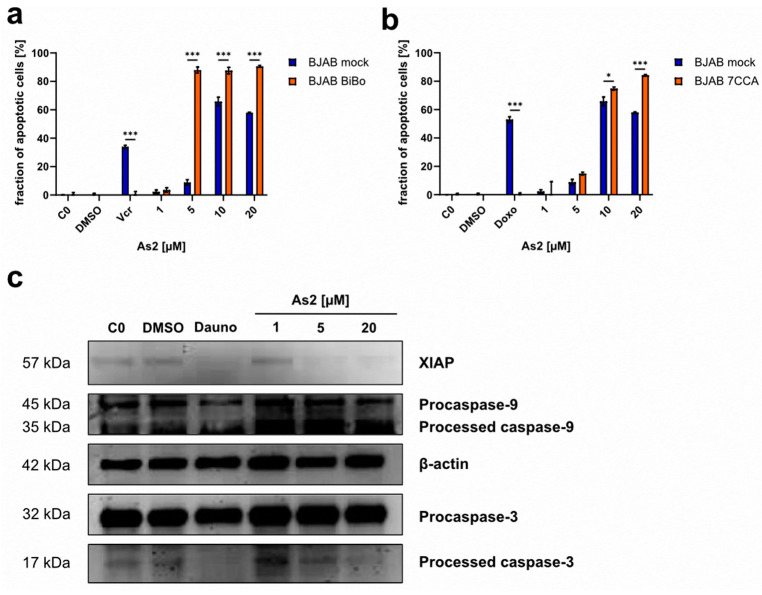
As2-induced apoptosis is not dependent on *Bcl-2* and caspase-3 expression levels. As2 treatment leads to caspase-9 and -3 processing and XIAP downregulation. Cell lines with established resistance to (**a**) Vincristine (Vcr; BJAB BiBo) and (**b**) Doxorubicin (Doxo; BJAB 7CCA) characterized by *Bcl-2* (B-cell lymphoma 2) overexpression and downregulation of caspase-3, respectively, and the non-resistant parental cell line BJAB mock without changes in *Bcl-2* and caspase-3 expression were treated with different concentrations of As2 [µM] or DMSO (dimethyl sulfoxide) and incubated for 72 h. Untreated negative controls (C0) were included as well as positive controls for drug resistance. The positive controls were treated with 5.6 nM of Vincristine (BJAB mock/BiBo) and 92 nM of Doxorubicin (BJAB mock/7CCA). The fraction of apoptotic cells in percentages was determined by flow cytometry, and the mean values with an SD of 3 replicates are depicted. Significant differences between cell types for each concentration were determined by a two-way analysis of variance (ANOVA) followed by Bonferroni’s multiple comparison test. Significance levels are determined as follows: (*) for *p* ≤ 0.05, and (***) for *p* ≤ 0.001. For simplicity, only significant differences are indicated. (**c**) Western blot analysis for the detection of the proteins XIAP and caspase-9 and -3 was performed with Nalm-6 cells that were treated with DMSO, 76 nM of Daunorubicin (Dauno), 1, 5, and 20 µM of As2, or left untreated (C0) for 36 h. β-actin served as a loading control. Protein band sizes are shown in kilodaltons (kDa).

**Figure 6 ijms-25-04713-f006:**
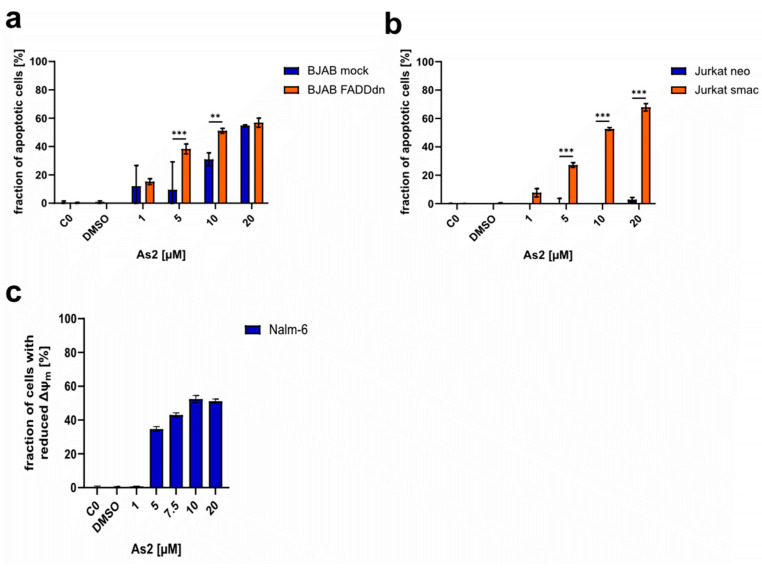
As2 induces an intrinsic apoptosis pathway through the mitochondrium. Cell lines characterized by (**a**) a dominant negative FADD mutant (BJAB FADDdn) and (**b**) overexpression of Smac/DIABLO (Jurkat smac) and respective control cell lines (BJAB mock and Jurkat neo) were treated with different concentrations of As2 [µM] or DMSO (dimethyl sulfoxide) and incubated for 72 h. Untreated negative controls (C0) were included. The fraction of apoptotic cells in percentages was determined by flow cytometry, and the mean values with an SD of 3 replicates are depicted. Significant differences between cell types for each concentration were determined by a two-way analysis of variance (ANOVA) followed by Bonferroni’s multiple comparison test. Significance levels are determined as follows: (**) for *p* ≤ 0.01, and (***) for *p* ≤ 0.001. For simplicity, only significant differences are indicated. (**c**) Using the JC-1 dye, the fraction of cells with reduced mitochondrial membrane potential (∆ψm) [%] was determined by flow cytometry after 48 h of incubation of Nalm-6 cells with DMSO, different concentrations of As2 [µM], or left untreated (C0). The mean values with an SD of 3 replicates are shown.

**Figure 7 ijms-25-04713-f007:**
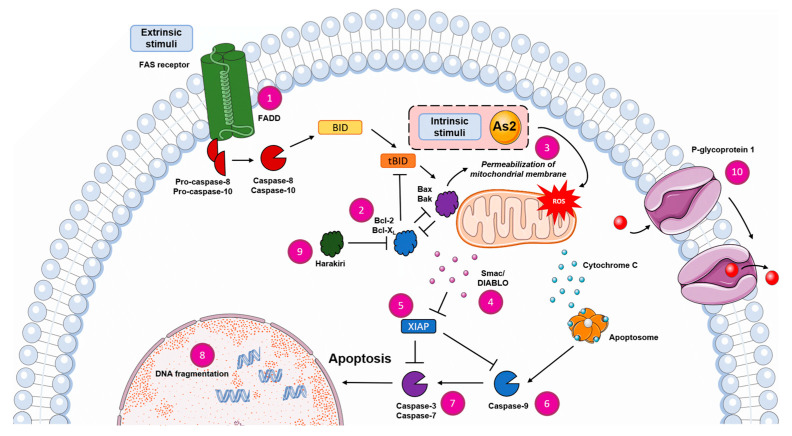
Schematic overview of the investigated factors of apoptosis induction and the potential role of As2 within the mitochondrial apoptosis pathway. While cell-extrinsic stimuli can activate the apoptosis pathway through cell surface receptors like the FAS receptor and lead to activation of caspase-8 and -10, intrinsic stimuli like DNA damage (e.g., through reactive oxygen species (ROS)) and potentially the compound As2 induce the intrinsic or mitochondrial apoptosis pathway. Both pathways result in DNA fragmentation and, eventually, cell death. The numbered factors in the apoptosis pathway were specifically investigated in this study: ① Apoptosis induction by extrinsic factors in BJAB FADDdn cells (Figure 6a), ② involvement of *Bcl-2* (B-cell lymphoma 2) in BJAB BiBo cells (Figure 5a), ③ changes in mitochondrial membrane potential using the JC-1 assay (Figure 6c), ④ contribution of Smac/DIABLO in apoptosis induction in Jurkat smac cells (Figure 6b), ⑤ XIAP (Figure 5c), ⑥ caspase-9 (Figure 5c), and ⑦ caspase-3 levels and activation through western blot analysis (Figure 5c), and ⑧ the eventual apoptosis induction by measurement of DNA fragmentation (Figure 1d). ⑨ Harakiri (Figure 3e) and ⑩ P-glycoprotein (Figure 3c,d) are involved in the resistance mechanisms of used cell lines (see Appendix A). The figure was generated using Servier Medical Art, provided by Servier, licensed under a Creative Commons Attribution 3.0 unported license.

**Table 1 ijms-25-04713-t001:** AC50 concentration of apoptosis induction in Nalm-6 cells by the 11 arsenic compounds As1–As11 depicted with their chemical structure.

Compound	Chemical Structure	AC50 [µM]
As1	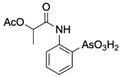	no effect
As2	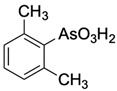	~6.3
As3	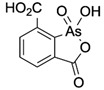	no effect
As4	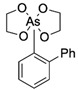	no effect
As5	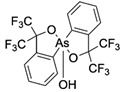	>100
As6	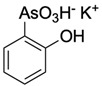	no effect
As7	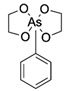	no effect
As8	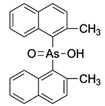	no effect
As9	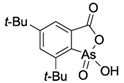	no effect
As10	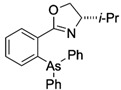	no effect
As11	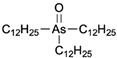	no effect

## Data Availability

The data presented in this study are available on reasonable request from the corresponding authors.

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
