# Peer review of "(2,6-Dimethylphenyl)arsonic Acid Induces Apoptosis through the Mitochondrial Pathway, Downregulates XIAP, and Overcomes Multidrug Resistance to Cytostatic Drugs in Leukemia and Lymphoma Cells In Vitro"

_ijms, 2024, doi:10.3390/ijms25094713_

Round 1
Reviewer 1 Report
Comments and Suggestions for Authors
The presented results of investigations on the anticancer potential of the new candidate based on an As-containing compound seem to be highly valuable. The development of new promising anticancer agents that would be able to overcome multidrug resistance is undoubtedly an urgent problem. The manuscript leaves a good impression in terms of the performed studies and achieved results. However, due to the fact most of the figures do not appear correctly (either due to the blakc background, the lack of axis labels, a,b,c designations, etc.), I cannot recommend its publication in the current form. Please provide the manuscript with correct figures.
Author Response
Thank you very much for taking the time to review our manuscript and for your comments on our research.
We appreciate that you pointed out the problems with the figure format. There seems to have been an error during the conversion of the figure formats. In the revised manuscript, we changed the figure format and tested whether the figures display correctly also after conversion of the manuscript to different file formats. We hope the issue is resolved now and all figures are displayed correctly in the updated version.
Reviewer 2 Report
Comments and Suggestions for Authors
-The abstract might be rephrased to make it more understandable. For example, simplifying complex phrases and expressing abbreviations.
-The introduction might be enhanced by providing a brief overview of the limitations or gaps in the existing utilization of arsenic-based therapies that the study seeks to tackle. provide a concise statement regarding the potential enhancements that As2 may offer in relation to the effectiveness and selectivity of current arsenic-based treatments.
- The concentration of seeded cancer cells in lines 435, 489, 505, and 519 should be10 powers of five, rather than 1x105 cells/mL.
Comments on the Quality of English Language
Minor editing of English language required. Most of the phrases are too long, which may reduce the quality of English and cause misunderstandings.
Author Response
Thank you very much for taking the time to review our manuscript. We appreciate your helpful comments and suggestions.
Below you can find our answers to each of your comments. In the revised manuscript, all changes are marked in red.
- Comment 1: "The abstract might be rephrased to make it more understandable. For example, simplifying complex phrases and expressing abbreviations."
- We rewrote the abstract to provide a more understandable overview, including the explanation of used abbreviations, while still adhering to the maximum word limit for the abstract.
- Comment 2: "The introduction might be enhanced by providing a brief overview of the limitations or gaps in the existing utilization of arsenic-based therapies that the study seeks to tackle. provide a concise statement regarding the potential enhancements that As2 may offer in relation to the effectiveness and selectivity of current arsenic-based treatments."
- We extended the introduction by including a more detailed review of current arsenic-based therapies, highlighting the need for new drug candidates like As2 that can overcome the limitations of existing therapy approaches.
- Comment 3: "The concentration of seeded cancer cells in lines 435, 489, 505, and 519 should be10 powers of five, rather than 1x105 cells/mL."
- We apologize for the formatting errors. Thank you for pointing them out. We corrected all wrongly formatted cell concentrations in the manuscript.
- Comment 4: "Minor editing of English language required. Most of the phrases are too long, which may reduce the quality of English and cause misunderstandings."
- Thank you for your suggestions for improving the readability of our manuscript. We rephrased parts of the manuscript, especially in the abstract, introduction and discussion, to avoid misunderstandings.
Reviewer 3 Report
Comments and Suggestions for Authors
The manuscript by Wilke et al. presents the antiproliferative effect of 2,6-dimethylphenyl)arsonic acid (As2) against leukemia and lymphoma cell lines. Strenghts of this work are the ability to overcome multidrug resistance, the identification of the mitochondrial apoptosis pathway as the main mode of action and the synergy with common anti-cancer drugs. The article is well organized and the relevant topic makes the article valid for publication in the journal. Nevertheless, severe lacks in the quality of presentation need to be addressed before publication. Hence, the following major revisions need to be done to improve the quality of the manuscript:
- The abstract is accurate and concise but the AC50 of As2 should be added.
- Introduction: some references are decade-old and should be replaced with more recent studies in the field. For example, instead of [4], the following articles could be mentioned concerning the search for novel potent anti-cancer drugs: doi: 10.1016/j.bmc.2023.117459; doi: 10.37349/etat.2022.00112; doi: 10.1158/0008-5472.CAN-22-1999; doi: 10.1016/j.ejmech.2023.115372.
- Furthermore, the use of arsenic compounds as anticancer agents should be deepened, adding more data about their efficacy and specifying for which type of cancers they are used.
- Paragraph 2.1: immediately specify that the synthesis of As1-11 was previously described.
- Almost all figures must be checked and corrected. Figure 1, 4 and 5 are unreadable because of the black background. Figures 3 and 6 are incomplete and the diagram axes are missing. These severe lacks in the style of presentation make it hard to analyse data and fully evaluate the manuscript.
- Line 82, corrects “1x105 cells/mL” in “1x105 cells/mL”
Comments on the Quality of English LanguageMinor editing of English language required
Author Response
Thank you very much for taking the time to review our manuscript and providing helpful comments on our research.
Below we address each of the comments and all changes in the revised manuscript are marked in red.
- Comment 1: "The abstract is accurate and concise but the AC50 of As2 should be added."
- Thank you for the suggestion. We added the AC50 of As2 to the abstract.
- Comment 2: "Introduction: some references are decade-old and should be replaced with more recent studies in the field."
- We added more recent publications on the topic of novel anti-cancer drug discoveries to the introduction. Thank you for pointing that out.
We would also like to mention, that in some specific cases, rather old publications were chosen in context of the more than 2000 years in which arsenic has been used as an therapeutic agent. - Comment 3: "Furthermore, the use of arsenic compounds as anticancer agents should be deepened, adding more data about their efficacy and specifying for which type of cancers they are used."
- In line with the previous comment, we also expanded on the current arsenic-based therapies, including the context of their use and potential limitations.
- Comment 4: " Paragraph 2.1: immediately specify that the synthesis of As1-11 was previously described."
- Thank you for the comment. We added the respective information and reference in the mentioned paragraph.
- Comment 5: "Almost all figures must be checked and corrected. Figure 1, 4 and 5 are unreadable because of the black background. Figures 3 and 6 are incomplete and the diagram axes are missing. These severe lacks in the style of presentation make it hard to analyse data and fully evaluate the manuscript."
- Thank you for pointing out this major formatting error. We replaced all figures with files that should now stay consistently readable throughout various format conversions.
- Comment 6: "Line 82, corrects “1x105 cells/mL” in “1x105 cells/mL”
- We corrected the mistakes for all wrongly formatted cell concentrations.
- Comment 7: "Minor editing of English language required"
- We rephrased several paragraphs, e.g. by shortening sentences in the abstract, introduction and discussion, to enhance readability and reduce the risk of misunderstandings.
Round 2
Reviewer 1 Report
Comments and Suggestions for Authors
The revised version of the mansucript can be accepted for publication.